# S-Scheme WO₃/SnIn₄S₈ Heterojunction for Water Purification: Enhanced Photocatalytic Performance and Mechanism

Pingfan Xu [1,2], Runqiu Zhang [1], Jiarong Gong [1], Yaofa Luo [1], Yihua Zhuang [1,*] and Peikun Zhang [1,*]

[1] School of Advanced manufacturing, Fuzhou University, Jinjiang 362200, China; xupingfan@fzu.edu.cn (P.X.); zrq980916@163.com (R.Z.); jiarong_gong@163.com (J.G.); luoyaofa@fzu.edu.cn (Y.L.)

[2] Key Laboratory of Ecological Environment and Information Atlas, Fujian Provincial University, Putian 351100, China

* Correspondence: yhzhuang@fzu.edu.cn (Y.Z.); pkzhang@fzu.edu.cn (P.Z.)

**Abstract:** Photocatalysis is a promising technology for removing micropollutants in water. However, developing efficient and stable catalysts remains a challenge. In this work, a novel step-scheme (S-scheme) heterojunction of $WO_3/SnIn_4S_8$ (WSI) was constructed through the combined process of in situ precipitation with hydrothermal synthesis to simultaneously realize photocatalytic degradation of bisphenol A(BPA) and reduction of Cr(VI) in contaminated water. Results showed that the WSI S-scheme heterojuction has a synergistic effect for the removal of BPA and Cr(VI). An optimum case of the WSI-12% heterojunction exhibited the highest photocatalytic efficiency in the degradation of BPA under visible light, which is ca. 2.5 and 3.8 times more than the pure $WO_3$ and SIS, respectively. The enhanced photocatalytic activity is attributed to the formation of the WSI S-scheme heterojunctions which facilitate the spatial separation of charge carriers and preserve strong photoredox ability. Further, the S-scheme mechanism of enhanced photocatalysis was examined by the radical-trapping experiment and ESR, and superoxide and hydroxyl radicals were determined to be the major reactive oxygen species responsible for BPA degradation and Cr(VI) reduction by WSI. This work provides a novel strategy for tailoring high-performance S-scheme heterojunctions and shows the promising application in purifying wastewater with complex pollutants.

**Keywords:** $WO_3/SnIn_4S_8$ heterostructure; bisphenol A; hexavalent chromium; photocatalytic degradation



## 1. Introduction

The increasing worldwide contamination of freshwater systems is a considerable environmental problem facing humanity in the twenty-first century. One cause of water quality degradation is micropollutants, including both inorganic and organic pollutants, such as toxic metal ions and endocrine-disrupting compounds (EDCs) in aquatic systems [1,2]. Of these micropollutants, hexavalent chromium (Cr(VI)) and bisphenol A (BPA) are the most common micropollutants found in industrial wastewater. Cr(VI) emitted from tanneries, electroplating and dyeing processes, is known to be toxic and carcinogenic and can cause health problems, such as liver damage, pulmonary congestions, vomiting, and severe diarrhea [3]. BPA, a raw material in the production of epoxy resin and polycarbonate plastic, may cause endocrine disruption in humans [4]. Although these pollutants are present at trace levels in effluents, their adverse effects on human health and aquatic ecosystems are extremely harmful, particularly when they are present in complex mixtures.

Wastewater treatment plants are usually designed to remove suspended particles and nutrient pollutants, while the overwhelming majority of micropollutants may be discharged directly into the environment or accumulated in sludge disposal [5,6]. Recent technologies including adsorption [7,8], membrane processes [9], electrochemical precipitation [10], and advanced oxidation processes [11], have been developed for the removal of micropollutants. Unfortunately, practical application of these technologies has suffered from several

limitations, such as high energy consumption, causing secondary pollution, and a narrow pH range for application. Therefore, the provision of economically and environmentally sustainable wastewater treatment for simultaneous removal of organic compounds and toxic metal ions remains a great challenge for wastewater treatment.

Heterogeneous photocatalysis is recognized as a promising technology for the simultaneous removal of organic pollutants and heavy metals. Over the past few decades, various types of single photocatalysts, such as $TiO_2$, $WO_3$, g-$C_3N_4$ and $MoS_2$, have been used for environmental remediation due to their efficient photocatalytic performance, high stability and environmental safety [12–15]. However, single photocatalysts cannot ensure both efficient light harvesting and strong redox ability. Another issue associated with single photocatalysts is the rapid recombination of charge pairs. To prevent these challenges, diverse strategies have been developed to enhance the photocatalytic activity of photocatalysts, including doping [16], metal loading [17], type II heterojunctions [18,19] and S-scheme heterojunctions [20]. Among them, the emerging S-scheme heterojunction, by combining two types of semiconductors, has been demonstrated as a promising strategy for energy conversion and environmental remediation, due to the effective spatial separation of photogenerated electron–hole pairs through the band alignment [21].

Generally, three design principles for S-scheme heterojunctions have been proposed in the previous literature. (1) An S-scheme heterojunction is composed of an oxidation photocatalyst (OP) and a reduction photocatalyst (RP). (2) The conduction band (CB) edge and Fermi level of RP should be higher than that of OP. (3) The work function of RP should be lower than that of OP. When a RP and an OP come into contact, the electron with a higher Fermi level in the RP drifts to the OP until their Fermi levels reach equilibrium. The redistribution of charge leads to the band bending and the formation of a built-in electric field, whose direction points from the RP to the OP. Upon light irradiation, the photogenerated electrons in the CB of the OP cross the heterojunction interface and recombine with the photogenerated holes in the VB of the RP under the action of the internal field. Ultimately, the photogenerated electrons of the RP and photogenerated holes of the OP can be preserved. Therefore, S-scheme heterojunctions have efficient spatial separation of charge carriers and maintain the strong photo-redox ability of both photogenerated electrons and holes.

An efficient S-scheme heterojunction for water purification requires the CB potential of the RP to be more negative than the reduction potential of oxygen to superoxide radicals ($\cdot O_2^-/O_2$ = −0.33 eV vs. NHE), while the VB position of the OP must be more positive than the oxidization potential of water to hydroxyl radicals ($\cdot OH/H_2O$ = +2.4 eV vs. NHE) [22,23]. Therefore, the accumulated holes and electrons are able to oxidize the $H_2O$ to $\cdot OH$ and reduce the $O_2$ to $\cdot O_2^-$. In order to design an S-scheme heterojunction, tungsten oxide ($WO_3$) is selected to couple with tin indium sulfide ($SnIn_4S_8$). $WO_3$, which has a band-gap energy of approximately 2.7 eV, can absorb visible photons and has a positive VB potential (ca. +3.1 eV vs. NHE), which can be used as an ideal OP that allows photoexcited holes to oxidize water into hydroxyl radicals [24,25]. $SnIn_4S_8$ (SIS) is another efficient visible-light photocatalyst with a relatively narrow band gap energy of approximately 2.0 eV [26]. The VB potential of SIS is located at approximately +1.30 eV vs. NHE, which is close to the CB position (+0.26 eV vs. NHE) of $WO_3$. After contact, the photogenerated electrons in the CB of $WO_3$ are inclined to recombine with holes in the VB position of SIS due to the Coulomb attraction. On the other hand, the CB potential of SIS is positioned at approximately −0.7 eV vs. NHE, which is more negative than the standard redox potential of $\cdot O_2^-/O_2$. Therefore, the photogenerated electron–hole pairs of $WO_3$/SIS S-scheme heterojunction are spatially separated in the two semiconductors, maintaining a strong redox potential.

Herein, a novel WSI S-scheme heterojunction was first fabricated via the combined process of in situ precipitation with hydrothermal synthesis to realize the perfect integration of the two semiconductors, as depicted in Scheme 1. Yellow $WO_3$ nanoparticles were synthesized by alcoholysis of $WCl_6$ followed by calcination of the blue $WO_3$ power. The as-

synthesized $WO_3$ were dispersed in water and metal ions, and $In^{3+}$ and $Sn^{4+}$ were deposited in advance on the $WO_3$ surface, where the metal ions can be adsorbed onto the surface through electrostatic interaction. Then, sodium diethyldithiocarbamate (DDTC-2Na) as sulfur source was added into the precursor solution to grow SIS nanosheets in situ onto the $WO_3$ surface, forming a direct S-scheme heterojunction. The crystal phase, morphology and optical property of the as-prepared samples were characterized by XRD, SEM, HRTEM, UV-vis DRS and PL spectrum. In addition, the simultaneous photocatalytic degradation of BPA and Cr(VI) reduction was conducted under visible light irradiation to evaluate the synergistic effect. Furthermore, the enhanced photocatalytic activity of the S-scheme mechanism was proposed based on the reactive oxygen species trapping experiments and the ESR characterizations. It was found that the WSI S-scheme heterojunction exhibits enhanced photocatalytic activity. This work provides a novel direct S-scheme system for efficient degradation of pollutant and potential application in both environmental protection and energy conversion.

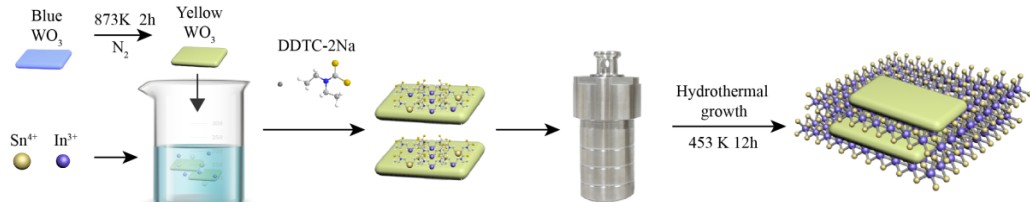

**Scheme 1.** Illustration of the fabrication of the WSI S-scheme heterojunction.

## 2. Results and Discussion

### 2.1. Structure and Morphology of WSI S-Scheme Heterojunction

The XRD patterns of the as-prepared samples showed a high degree of crystallinity, as shown in Figure 1. The distinct peaks of bare SIS at 14.3°, 23.4°, 27.5°, 33.3°, 43.7°, and 47.8° could be ascribed to (111), (220), (311), (400), (333), and (440) crystal planes of cubic SIS (JCPDS card # 42-1305, a = b = c = 10.7507 Å), which is consistent with a previous report on the crystal planes of cubic SIS [26]. In addition, the XRD patterns of the pure $WO_3$ are perfectly ascribed to the monoclinic phase of $WO_3$ (JCPDS card # 43-1035, a = 7.297 Å, b = 7.539 Å, c = 7.688 Å), which exhibits higher photocatalytic activity and is more stable than the orthorhombic and hexagonal phase [13]. Generally, the intensity of diffraction peaks is proportional to the amount of each phase present. In the present study, the characteristic peaks of WSI heterojunction are well consistent with the pattern of pure SIS. Compared with the peaks of SIS, the peak intensity of $WO_3$ is relatively weak owing to the low proportion of $WO_3$ in the heterojunction. No characteristic peaks of other impurities were observed, indicating that the WSI heterojunction was successfully prepared. Importantly, all diffraction peaks of SIS remain unchanged in their position and intensity after integration with $WO_3$ via hydrothermal synthesis, demonstrating that the introduction of $WO_3$ does not alter the crystal structure of SIS.

The morphologies of the as-prepared samples were observed by SEM and HRTEM. The SEM image shows that the as-synthesized $WO_3$ present a flat sheet-like structure (Figure 2a), which is similar to the previous report [27]. As shown in Figure 2b, SIS was grown in situ onto a $WO_3$ surface via hydrothermal synthesis to form a layered structure of WSI-12% heterojunction, indicating that a $WO_3$ nanosheet was embedded in SIS and the S-scheme heterojunction was successfully fabricated. The HRTEM image of the WSI-12% heterojunction was analyzed to further investigate the morphology and crystal structure. As shown in Figure 2c, the $WO_3$ nanosheet was wrapped uniformly through in situ synthesis of SIS, indicating that the direct S-scheme heterojunction was successfully synthesized. The intimate contact between $WO_3$ and SIS could promote electron transfer from $WO_3$ to SIS under visible light irradiation, which is the key factor for the successful construction of the S-scheme heterojunction. In addition, the lattice fringe of $WO_3$ and SIS was measured to further investigate the detailed information of the crystalline structure. As shown in

Figure 2d, the lattice spacing of 0.52 nm, 0.384 nm, and 0.269 nm can be, respectively, assigned to the (110), (002), and (002) planes of $WO_3$, which agrees with the XRD analysis. For the SIS nanosheet, the obvious lattice spacing values were 0.267 nm and 0.62 nm, which match the distance between the (400) and (111) lattice planes of the cubic SIS structure, respectively. It should be noted that the $WO_3$ nanosheets were stuck in the middle of SIS, as shown in Figure 2e,f. The obvious layered structure indicates that the direct S-scheme heterostructure is successfully synthesized through a combined method of in situ precipitation with the hydrothermal synthesis.

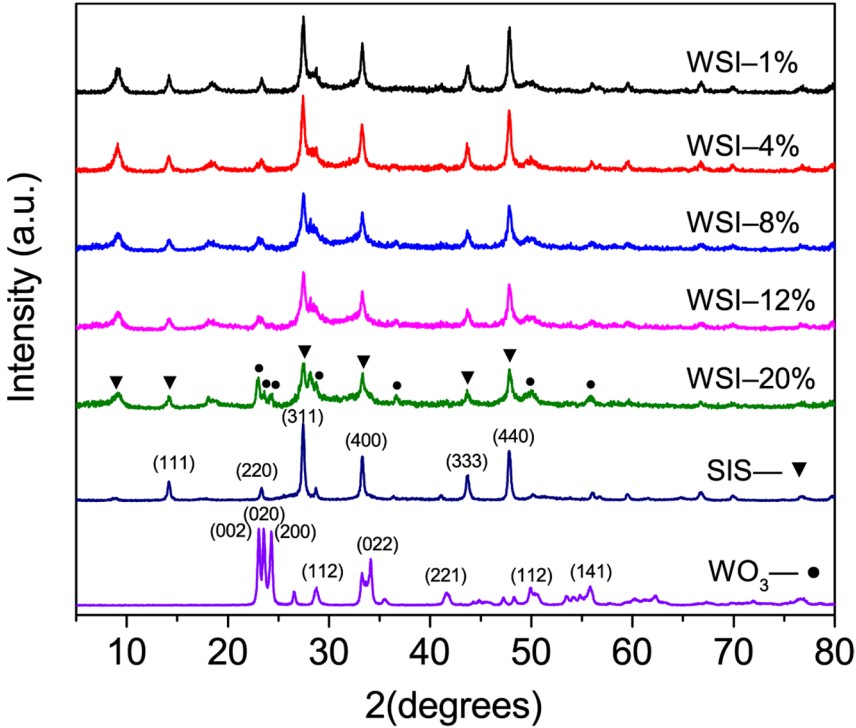

**Figure 1.** XRD patterns of SIS, $WO_3$, and WSI S-scheme heterojunction.

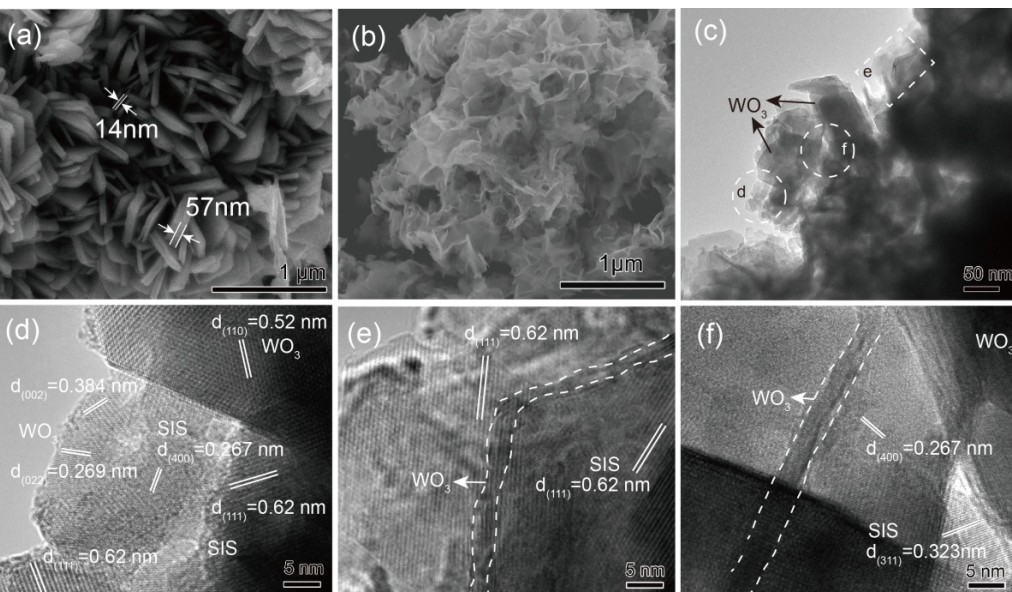

**Figure 2.** SEM images of (**a**) $WO_3$ and (**b**) WSI photocatalyst, (**c**–**f**) HRTEM image of WSI-12% photocatalyst.

The chemical states of SIS and the WSI-12% heterostructure were investigated by XPS analysis, as shown in Figure 3. The high-resolution Sn 3d and In 3d spectrum exhibit two peaks, 3d5/2 and 3d3/2, with spin-orbital splittings of 8.5 eV and 7.6 eV, respectively, indicating that the valence states in the composite are $Sn^{4+}$ and $In^{3+}$ [28,29]. The binding energy of Sn 3d in the WSI-12% heterostructure decreases compared with SIS, due to the electronegativity of oxygen atoms boned to $Sn^{4+}$ after integration of $WO_3$. Similarly, the binding energy of In 3d in the WSI-12% heterostructure shifts from 444.5 eV to 444.4 eV under the influence of oxygen atoms in $WO_3$, as observed in $ZnIn_2S_4/TiO_2$, where the binding energy of In 3d is influenced by $TiO_2$ [30]. The S 2p spectrum shows a broad spectrum with a hump at higher binding energy, which is divided into two peaks, assigned to 2p3/2 and 2p1/2, respectively. The spin-orbital splitting of S 2p in the composites is 1.1 eV, indicating that the oxidation state of S is −2. Compared to S 2p in SIS and WSI-12%, the decrease of the binding energy is obvious under the influence of the W and O atoms, due to the increase in electron density of the S atom. Moreover, the peak area ratios and the spin-orbital splitting are 1.3 and 2.15 eV, respectively, indicating that the chemical state of W 4f in the composite is $W^{6+}$ [31].

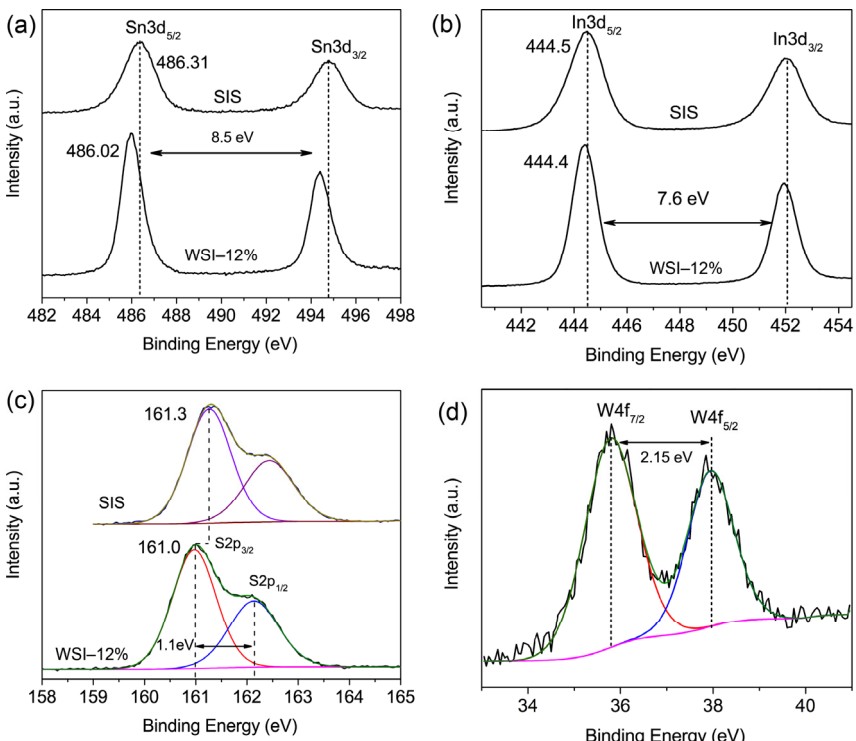

**Figure 3.** XPS spectrum of SIS and WSI-12% photocatalyst: (**a**) Sn 3d, (**b**) In 3d, (**c**) S 2p, (**d**) W 4f.

### 2.2. Optical Properties and Charge Separation Efficiency of WSI S-Scheme Heterojunction

The UV-visible diffuse reflectance spectra of pure SIS, $WO_3$ and the WSI heterojunction are shown in Figure 4a,b. Both the SIS and $WO_3$ semiconductors have strong absorption in the visible light because of their narrow band gaps. The visible light absorption property of WSI-12% is between that of SIS and $WO_3$, with a maximum absorption wavelength larger than 608 nm. The optical band gaps of SIS, WSI (1%, 4%, 8%, 12%, 20%) and $WO_3$ was estimated from the transmittance data using Tauc's plot ($\alpha h v \approx (h v\text{-}E g)^{1/2}$) [32], which are 2.08, 2.17, 2.21, 2.26, 2.24, 2.29, and 2.83 eV, respectively. The band gap of the WSI heterojunction is located between that of SIS and $WO_3$, but the value does not increase proportionally with increasing $WO_3$ doping amount, indicating that the WSI heterojunction is not simply mixed.

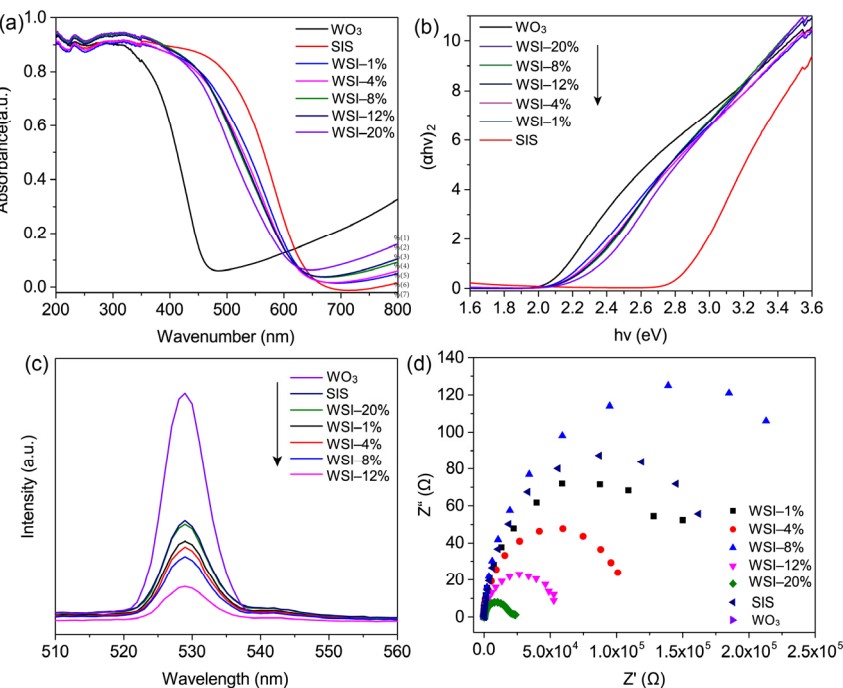

**Figure 4.** (**a**) DRS spectra of the as-prepared sample; (**b**) Plots of the $(\alpha h v)^2$ vs. photo energy (*hv*) for the as-prepared sample; (**c**) PL emission spectra ($\lambda_{ex}$ = 350 nm) and (**d**) EIS spectra of the as-prepared sample. The direction of the arrows corresponds to the spectra of different samples.

The photocatalytic activity of a photocatalyst mainly depends on the separation of photogenerated carriers. The PL spectrum was utilized to investigate the separation efficiency of photogenerated electron–hole pairs. Generally, lower PL emission intensity indicates that the recombination rate of photogenerated electron–hole pairs is low, thus resulting in better photocatalytic activity [33]. As shown in Figure 4c, the PL intensity of $WO_3$ and SIS was stronger than that of the WSI heterojunction. The minimum PL intensity is ascribed to the WSI-12% heterojunction, suggesting that the construction of the S-scheme heterojunction facilitates the separation of photogenerated electron–hole pairs, which is expected to enhance photocatalytic activity.

Electrochemical Impedance Spectroscopy (EIS) of the photocatalysts was measured to further investigate the photogenerated electron transfer efficiency. It is reported that the smaller arc radius implies better conductivity and higher electron transfer efficiency [34]. As shown in Figure 4d, WSI-12% and WSI-20% have a relatively small arc radius compared with the bare $WO_3$ and SIS, indicating the highest conductivity and the fastest electron transfer efficiency are under visible light irradiation.

### 2.3. Photocatalytic Activity of WSI S-Scheme Heterojunction

The photocatalytic performance of the photocatalysts was evaluated by the degradation of BPA and reduction of Cr(VI), respectively, under visible light irradiation. As shown in Figure 5a, the photocatalytic efficiency of the WSI heterojunction is significantly higher than that of pure SIS and $WO_3$. Moreover, the photocatalytic degradation rate was calculated to evaluate the photocatalytic activity. The photocatalytic degradation rate fitted well with the pseudo-first-order kinetics. As shown in Table 1, the photocatalytic rate constant of BPA degradation by WSI-12% was estimated to be 0.2135 $h^{-1}$, which was 2.5 and 3.8 times that of $WO_3$ and SIS, respectively. For the reduction of Cr(VI), the apparent rate constant of WSI-12% was 6.5 and 1.5 times that of $WO_3$ and SIS, respectively. The enhanced photocatalytic activity can be attributed to the favorable band alignment between $WO_3$ and SIS, which forms a direct S-scheme heterojunction that greatly improves the separation of charge carriers in the two-phase interface. Hence, the construction of the S-

scheme heterojunction shows great potential as an efficient photocatalyst for simultaneous oxidation of BPA and reduction of Cr(VI).

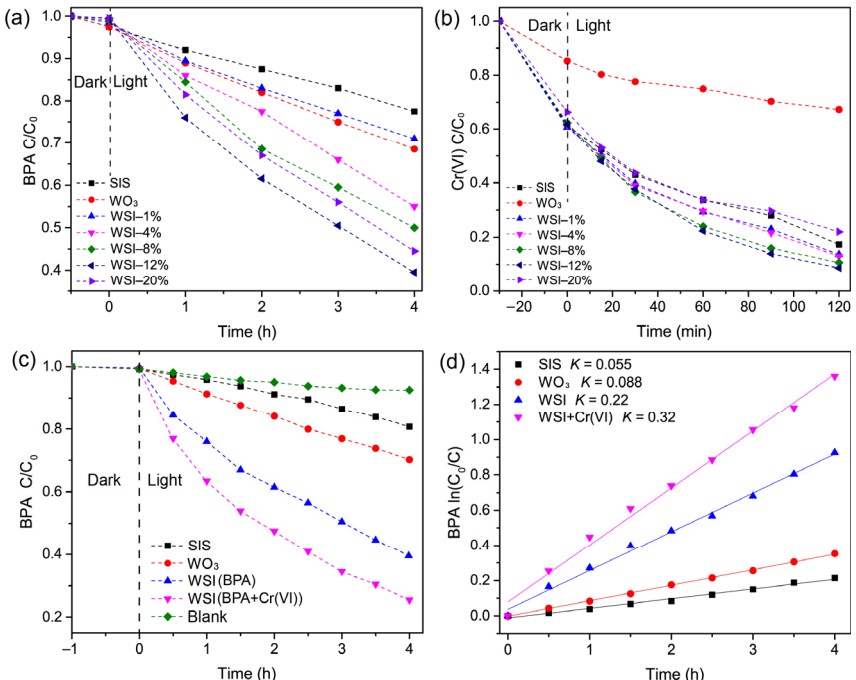

**Figure 5.** Photocatalytic performance of the as-prepared sample: (**a**) BPA oxidation, (**b**) Cr(VI) reduction; simultaneous photocatalytic oxidation of BPA and reduction of Cr(VI) over WO$_3$, SIS and WSI-12% heterojunction under visible irradiation, (**c**) photocatalytic efficiency, (**d**) photocatalytic degradation rate constant.

**Table 1.** Kinetic reaction rate constants for the degradation of BPA and reduction of Cr(VI).

| Photocatalyst | Reaction Rate Constants ($k$)/min$^{-1}$ | |
| :---: | :---: | :---: |
| | **BPA** | **Cr(VI)** |
| SIS | 0.0557 | 0.6432 |
| WO$_3$ | 0.0852 | 0.1494 |
| WSI-1% | 0.0780 | 0.7482 |
| WSI-4% | 0.1339 | 0.7728 |
| WSI-8% | 0.1614 | 0.8970 |
| WSI-12% | 0.2135 | 0.9834 |
| WSI-20% | 0.1853 | 0.5772 |

The simultaneous photocatalytic degradation of BPA and reduction of Cr(VI) with the WSI-12% heterojunction was studied to evaluate the S-scheme mechanism. Figure 5c,d show the BPA degradation efficiency and the corresponding rate constant in coexistence with Cr(VI). No significant degradation of BPA was detected in the dark without the photocatalyst, indicating that no reaction occurred between Cr(VI) and BPA. The degradation efficiency of BPA with and without Cr(VI) was 61.5% and 74.6%, respectively, which demonstrated that the coexistence of Cr(VI) could significantly enhance the BPA degradation efficiency. Moreover, the degradation process also fits well with the pseudo-first-order kinetics model when BPA is mixed with Cr(VI). As shown in Figure 5d, the degradation rate constant of BPA/Cr(VI) matrix is 1.5 times more than that observed with BPA alone. The photocatalytic reduction efficiency of Cr(VI) in the absence/presence of BPA is 85.9% and 96.9%, respectively, after 2 h of visible light irradiation. At the same time, the reduction rate constant of Cr(VI) in the presence of BPA is 1.75 times that of Cr(VI) alone. According

to the basic principle of photocatalysis, the separation of photogenerated electrons and holes is accelerated in the presence of photogenerated electrons and hole traps. In this case, BPA and Cr(VI) were used as the electron donor and electron acceptor, respectively. Hence, the coexistence of Cr(VI) could accelerate the oxidation of BPA and vice versa. BPA was degraded by photogenerated holes and the corresponding hydroxyl radical produced by VB position of $WO_3$, and Cr(VI) as electron acceptor is directly reduced by photogenerated electrons produced by the CB of SIS, resulting in the efficient separation of photogenerated electron–hole pairs in the WSI heterojunction. As a result, there are synergy effects that mutually promote the degradation efficiency with the coexistence of BPA and Cr(VI).

### 2.4. Reaction Mechanism

Photocatalytic Degradation Intermediates and Pathway

Three-dimensional excitation emission matrix fluorescence spectroscopy (3D EEM) was used to identify the trace intermediates generated in the photocatalytic degradation of BPA. According to the previous report [35], the delocalized electrons in the conjugate system of benzene and its derivatives were easily excited to produce fluorescence signals. In this case, the degradation intermediates of BPA, including benzene, were determined by 3D EMMs to preliminarily estimate the pathway of photodegradation. As shown in Figure 6a, no obvious fluorescence spectrum peak was detected in ultrapure water, which served as a blank sample. All the fluorescence spectrum samples were subtracted to eliminate the Rayleigh scattering. As shown in Figure 6b, the three-dimensional EEM spectroscopy of a BPA sample displayed two weak fluorescence signals located at $\lambda_{ex}/\lambda_{em}$ = 273/305 nm and $\lambda_{ex}/\lambda_{em}$ = 226/305 nm [36]. The emission fluorescence peak of BPA disappeared under visible irradiation for one hour. Meanwhile, two new fluorescence signals located at $\lambda_{ex}/\lambda_{em}$ = 220/300 nm and $\lambda_{ex}/\lambda_{em}$ = 260/353 nm appeared, which were attributed to the phenol fluorescence signal and the low-molecular-weight organic matter or organic matter containing carboxyl groups [37]. After 2 h of irradiation, the above fluorescence peaks were significantly enhanced, indicating an increase in the concentration of intermediates. Meanwhile, the fluorescence peak located at $\lambda_{ex}/\lambda_{em}$ = 275/300 nm appeared, which was ascribed to isopropylated phenyl. The intensity of the fluorescence peak became weak after 3 h of photocatalytic degradation, indicating that the low-molecular-weight organic matter was mineralized. After 4 h of irradiation, all fluorescence signals expected for the signal phenol and low-molecular organic matter disappeared, indicating that BPA was decomposed into phenol and low-molecular organic matter.

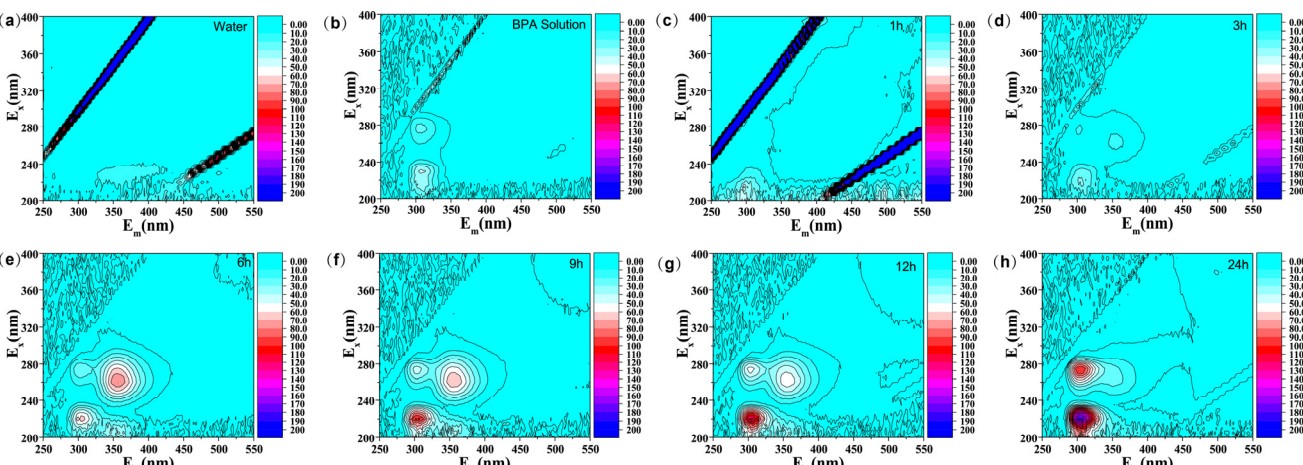

**Figure 6.** Three-dimensional fluorescence spectra of BPA-degraded production at different illumination times: (**a**) Blank–0 h; (**b**) Original solution–0 h; (**c**) 1 h; (**d**) 3 h; (**e**) 6 hours; (**f**) 9 h; (**g**) 12 h; (**h**) 24 h.

To further deepen our understanding of the photocatalytic degradation process, GC/MS was conducted to identify the degradation intermediates of BPA after 4 h of

irradiation. Figure S2 displays the MS spectra of the intermediates observed during the photocatalytic oxidation reaction. Table S1 summarizes the main fragment ions (*m/z*) and abundance (%) of reaction products in comparison to the NIST data library. As shown in Figure S1 and Table S1, the chromatogram and mass spectrum of the BPA degraded intermediate were listed in the order according to the retention time. The intermediates were as follows: 4-hydroxy butyric acid(B), phenol(C), benzoquinone(D), 2-dipropylmalonic acid©, benzene acetaldehyde(F), hydroquinone(G), Penta-1,4-dien-3-one(H), 4-isopropenylphenol(I), 4-isopropylphenol(J) and p-Hydroxyacetophenone (K). Based on the detected intermediates and relevant literature reports, three possible degradation pathways of BPA were proposed, as illustrated in Figure 7 [38,39]. The first pathway is initiated by the cleavage of the C-C bond between phenyl groups and the central carbon atom and leads to the formation of a phenol radical and an isopropylphenol radical. In Pathway 1, hydroxyl radicals (·OH) attack the C-C bond between phenyl groups and the central carbon atom, leading to the formation of the phenol radical and isopropylphenol radical. Some of the phenol radicals can further oxidize to produce hydroquinone (G) and benzoquinone (D), while another portion of the phenol radicals is quenched to form phenol (C). The isopropylphenol radical undergoes additional transformation, resulting in the formation of 4-isopropenylphenol (J) and 4-isopropylphenol (I), with the latter being oxidized to p-Hydroxyacetophenone. The phenol products continue to oxidize, breaking the benzene ring and forming smaller molecules, such as 4-hydroxybutyric acid, Penta-1,4-dien-3-one, ultimately leading to their complete oxidation into $CO_2$ and $H_2O$. Another potential degradation pathway involves the direct oxidation and demethylation of BPA, resulting in the formation of phenol (C), benzene acetaldehyde (F), and p-Hydroxyacetophenone (K), which subsequently degrade into low-molecular-weight organic acids. The hydroxyl radical is the major active radical involved in decomposing the benzene ring due to its strong oxidation properties. In this work, we demonstrated that the redox potential of photogenerated holes was enhanced by constructing the WSI heterojunction and that BPA could be oxidized by ·OH and mineralized into low-molecular-weight organics.

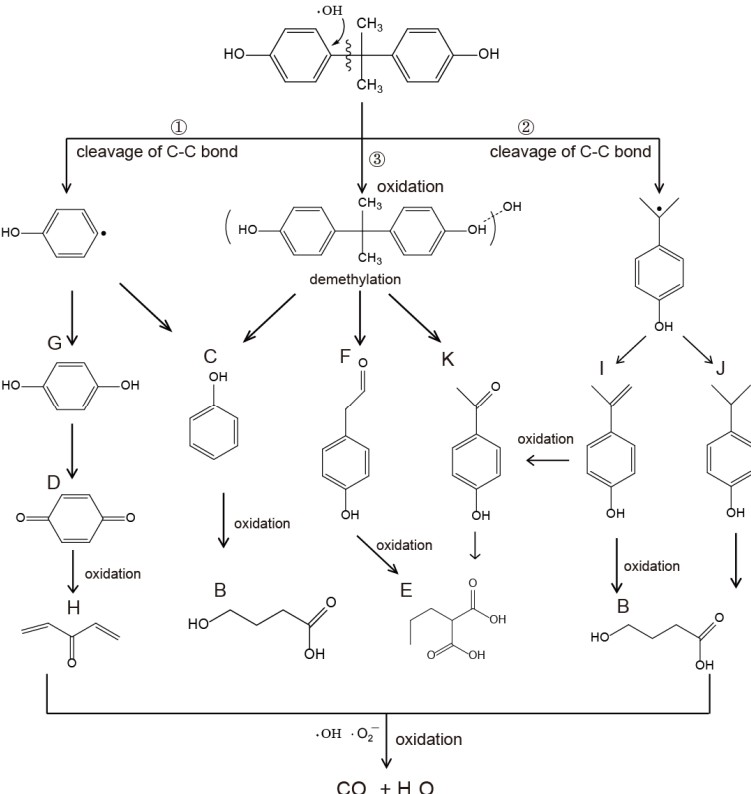

**Figure 7.** Possible degradation pathway of BPA over WSI-12% heterojunction.

### 2.5. S-Scheme Mechanism

The study investigated the reactive species of the photocatalytic reaction to explore the S-scheme mechanism of the WSI heterojunction. Initially, benzoquinone (BQ), isopropyl alcohol (IPA) and ammonium oxalate (AO) were selected to trap the radical of $\cdot O_2^-$, h+ and $\cdot OH$, respectively, in the photocatalytic degradation of BPA. As shown in Figure 8a, the photocatalytic degradation efficiency of BPA was inhibited in the presence of traps. The degradation efficiency was 57.9%, 63.3% and 84.5% when adding the trap of BQ, IPA, and AO, respectively, while the degradation efficiency was 91.7% in the absence of traps, indicating that the major active species during BPA degradation were $\cdot O_2^-$ and $\cdot OH$. The minor inhibition of degraded BPA efficiency in the presence of AO suggested that the radical of h+ contributed little to the dye photodegradation. The role of the reactive species was as follows: $\cdot O_2^- > \cdot OH > h^+$.

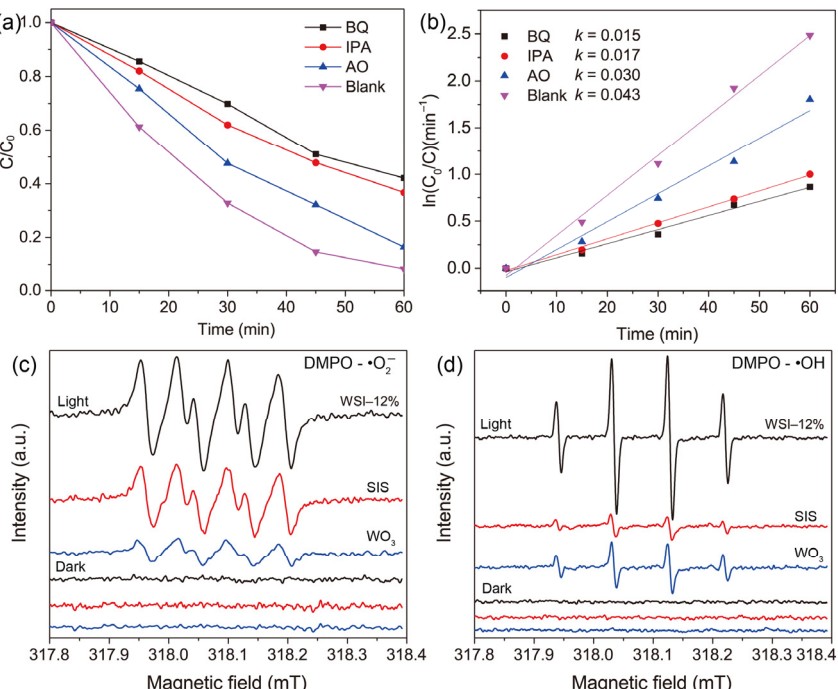

**Figure 8.** Effect of scavengers on the degradation of BPA over WSI-12%: (**a**) photocatalytic efficiency and (**b**) degradation rate, ESR signals of the (**c**) DMPO-$\cdot O_2^-$ and (**d**) DMPO-$\cdot OH$ of WSI-12% photocatalyst.

To further investigate the concentration of reactive oxygen species in the photodegradation of organic compounds, the ESR spin-trap technique (with DMPO) was employed to measure the signals of DMPO-$\cdot O_2^-$ and DMPO-$\cdot OH$. The ESR signals of DMPO-$\cdot O_2^-$ and DMPO-$\cdot OH$ were shown in Figure 8c,d. No obvious signals were detected in the dark reaction process. In contrast, the ESR signals of DMPO-$\cdot O_2^-$ and DMPO-$\cdot OH$ appeared after 10 min of irradiation. Both the DMPO-$\cdot O_2^-$ and DMPO-$\cdot OH$ signals were significantly enhanced in the WSI-12% heterojunction compared with the single-component photocatalyst, indicating that the concentrations of $\cdot O_2^-$ and $\cdot OH$ radicals were higher in the WSI-12% reaction system. The ESR signals of DMPO-$\cdot OH$ produced by WO$_3$ were relatively strong, which indicated that the $\cdot OH$ radical played a more important role in the degradation of organic compounds. As for the SIS, the $\cdot O_2^-$ radical contributed significantly to the photocatalytic degradation based on the ESR signals of DMPO-$\cdot O_2^-$. According the ESR results, it is further confirmed that the efficient separation and transfer of photogenerated electron–hole pairs were improved by the construction of the heterojunction. In addition, the redox ability of the photogenerated hole and electron was enhanced, resulting in improved photocatalytic activity.

Based on the above results and analysis, a possible S-scheme mechanism of WSI-12% was proposed, as shown in Figure 9. The CB and VB redox potentials of $WO_3$ were +0.13 V vs. NHE and +2.96 V vs. NHE, respectively. The SIS photocatalyst, with a redox potential of $-0.64$ V vs. NHE (CB) and +1.44 V vs. NHE (VB), could couple with $WO_3$ to form the direct S-scheme system, due to the more negative CB potential. The redox potential between the VB of SIS and the CB of $WO_3$ is 1.31 V, which facilitates the transfer of the photogenerated electron to the VB position of SIS. Under visible light irradiation, the photogenerated electron of the WSI-12% S-scheme system transfers from the CB of $WO_3$ to the VB of SIS and recombines with the hole of SIS. Therefore, the electron on the CB position of SIS and the hole on the VB position of $WO_3$ are spatially separated, increasing the probability of reaction with reactants. The VB potential of $WO_3$ is larger than the oxidization potential of water to hydroxyl radicals ($\cdot OH/H_2O$ = 2.4 eV vs. NHE), while the CB potential of SIS is more negative than the reduction potential of oxygen to superoxide radicals ($\cdot O_2^-/O_2$ = $-0.33$ eV vs. NHE). The $\cdot OH$ and $\cdot O_2$- radical generated by the photo-introduced holes and electrons of the WSI S-scheme system can oxidize the refractory organics to low-molecular organics, and the photogenerated electrons can reduce the Cr(VI) to Cr(III). Hence, the S-scheme system can simultaneous oxidize organic and reduce heavy metal ions. The major reaction steps of the S-scheme mechanism under visible light irradiation are summarized in the following reactions:

$$WSI \xrightarrow{h\nu} SIS(e^-/h^+)/WO_3(e^-/h^+) \tag{1}$$

$$SIS(e^-/h^+)/WO_3(e^-/h^+) \rightarrow SIS(e^-) + WO_3(h^+) \tag{2}$$

$$SIS(e^-) + O_2 \rightarrow O_2^- \cdot \tag{3}$$

$$WO_3(h^+) + H_2O \rightarrow \cdot OH \tag{4}$$

$$6SIS(e^-) + Cr_2O_7^{2-} + 14H^+ \rightarrow 2Cr^{3+} + 7H_2O \tag{5}$$

$$\cdot O_2^- / \cdot OH + BPA \rightarrow \cdots \rightarrow CO_2 + H_2O \tag{6}$$

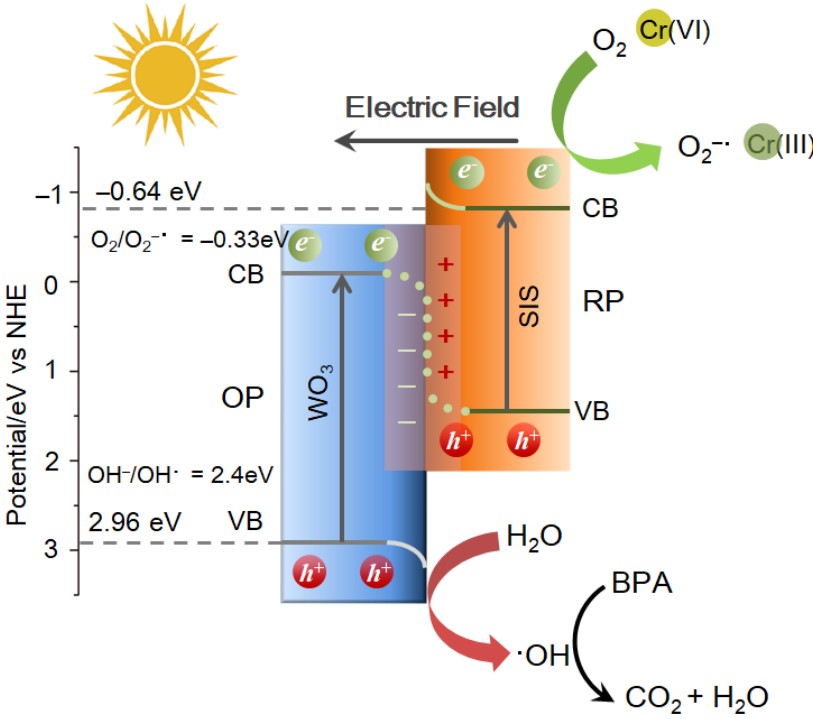

**Figure 9.** S-scheme transfer mechanism of photogenerated electrons under visible light irradiation.

## 3. Materials and Methods

### 3.1. Synthesis of WSI S-Scheme Heterojunction

Yellow $WO_3$ nanosheets were synthesized according to a previous report via a solvothermal alcoholysis process followed by calcination of the hierarchical structure of the as-prepared $WO_3$ [27]. Furthermore, a direct WSI S-scheme heterojunction was fabricated by combining the in situ precipitation and hydrothermal synthesis method. In brief, $WO_3$ nanosheets were dispersed in deionized water and ultrasonized for 10 min. Meanwhile, $SnCl_4 \cdot 5H_2O$ and $InCl_3 \cdot 4H_2O$ were dissolved in aqueous acetic acid solution, and the molar ratio of $Sn^{4+}$ and $In^{3+}$ was 1:4. The concentration of the $Sn^{4+}/In^{3+}$ mixture was 0.3 mM. Then, the above solution was mixed with the $WO_3$ suspension, and 0.24 mM DDTC-2Na was dropwise added to the mixture. The suspension was vigorously stirred for 2 h and then introduced into a 50 mL Teflon-lined autoclave and maintained at 453 K for 12 h. After cooling down to room temperature, the yellow powder was collected by filtration and repeatedly washed by deionized water and ethanol, finally freeze-dried to obtain the WSI heterojunction. Bare $WO_3$ and SIS were also prepared under the same conditions. For the WSI heterojunction, the mass percentage of $WO_3$ to SIS ranged from 1% to 20%. The as-prepared composites were labeled as WSI-1%, WSI-4%, WSI-8%, WSI-12%, and WGS-20%, respectively.

### 3.2. Characterization of WSI S-Scheme Heterojunction

X-ray diffraction (XRD) patterns were obtained using a Miniflex600 XRD spectroscopy instrument (Rigaku, Tokyo, Japan) with Ni-filtered Cu Kα irradiation (λ = 1.5406 Å) at a scanning rate of 2°/min, ranging from 10° to 80°. The structures and morphologies of the samples were characterized using field emission scanning electron microscopy (FESEM, FEI, Nova Nano SEM450, Hillsboro, OR, USA) and a high-resolution transmission electronic micrograph (HRTEM, FEI, Tecnai G220, Hillsboro, OR, USA). The chemical bonding states of samples were investigated using X-ray photoelectron spectroscopy (XPS, ESCA Lab250, Waltham, MA, USA) equipped with twin anode Al Kα radiation, and the binding energies were calibrated by referencing the C 1s (284.6 eV) peak. The optical properties were measured by a UV-Vis spectrophotometer (Cary 5000, Agilent, Santa Clara, CA, USA) and a photoluminescence spectrum (PL, Cary Eclipse, Agilent). EIS measurements were conducted using the EIS system (EIS, CHI660, CH instruments, Beijing, China) in a frequency range of $0.1–10^5$ Hz, utilizing an initial voltage of 0 V and amplitude of 5 mV. The photocatalytic degradation intermediates of BPA were identified using gas chromatography–mass spectrometry (GC/MS, 7890B/5977, Agilent, Santa Clara, CA, USA). Finally, the reactive oxygen species were verified using electron spin resonance (ESR, JES-FA200, JESO, Okinawa, Japan).

### 3.3. Photocatalytic Experiment

The photocatalytic experiments were performed in a double-walled quartz jacket filled with cool water under visible light irradiation (Figure S3). A 500W Xenon lamp coupled with a cutoff filter (λ > 420 nm) was used as the light source. The distance between Xe lamp and reactor was adjusted to 10 cm to achieve an incident-light intensity of 100 mW·cm$^{-2}$ (equivalent to one sun). First, the photocatalytic activity of the WSI heterojunction was evaluated by photocatalytic degradation of BPA and reduction of Cr(VI), respectively. Subsequently, simultaneous photocatalytic oxidation of BPA and reduction of Cr(VI) was conducted to investigate the synergetic decontamination effects. Briefly, 20 mg of photocatalyst was dispersed in 50 mL aqueous solution with the initial concentration ($C_0$) of 20 mg·L$^{-1}$ Cr(VI), 20 mg·L$^{-1}$ BPA and a mixture of Cr(VI) and BPA. The pH of the solution was adjusted using diluted NaOH or HCl. Prior to irradiation, the suspension was magnetically stirred for 30 min in the dark to achieve an adsorption equilibrium between reactants and photocatalysts. During irradiation, 3 mL of suspension was sampled at specific time intervals and centrifuged to remove the photocatalysts. The concentration of Cr(VI) and BPA was measured using UV-Vis spectrophotometry and

high-performance liquid chromatography (HPLC) with a UV detector. The photocatalytic efficiency was calculated according to $C/C_0$, where $C_0$ was the equilibrium concentration after adsorption and $C$ was the concentration of Cr(VI) and BPA after the reaction.

## 4. Conclusions

In this work, a novel direct S-scheme photocatalyst was prepared by combining in situ precipitation with hydrothermal synthesis to achieve perfect integration of $WO_3$ and SIS. The construction of the S-scheme heterojunction promoted the separation of photogenerated charges while maintaining the strong redox ability, leading to its excellent photocatalytic activity towards the oxidation of BPA. Moreover, it exhibited outstanding performance for simultaneous photocatalytic degradation of BPA and reduction of Cr(VI). BPA was decomposed into low-molecular-weight organic compounds and finally mineralized into $CO_2$ and $H_2O$. Radical trap and ESR experiments indicated that the major active species in the process of BPA degradation were $\cdot O_2^-$ and $\cdot OH$. This study provides a novel strategy for designing and constructing a direct S-scheme photocatalyst for highly efficient decontamination of organic pollutants and heavy metal ions.

**Supplementary Materials:** The following supporting information can be downloaded at: https://www.mdpi.com/article/10.3390/catal13111450/s1, Figure S1: GC chromatogram of BPA after 4 h degradation; Figure S2: Mass spectrum of the intermediate products of BPA after 4 h degradation; Figure S3: Schematic diagram of photocatalytic reactor; Table S1: GC-MS data of the photodegradation products of BPA by WSI.

**Author Contributions:** P.X.: Conceptualization, Methodology, Writing—Original draft preparation. R.Z.: Investigation, Data Curation. J.G.: Investigation, Formal analysis. Y.L.: Investigation, Formal analysis. Y.Z.: Resources, Writing—Reviewing and Editing. P.Z.: Conceptualization, Writing—Reviewing and Editing. All authors have read and agreed to the published version of the manuscript.

**Funding:** This research was funded by Key Laboratory of Ecological Environment and Information Atlas (grant number ST21001), and Technology Innovation Center for Exploitation of Marine Biological Resources (TIRMBR202104). And The APC was funded by Technology Innovation Center for Exploitation of Marine Biological Resources.

**Data Availability Statement:** The data presented in this study are available in the article and its online supplementary information.

**Conflicts of Interest:** The authors declare no conflict of interest.

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
