# Peer review of "S-Scheme WO3/SnIn4S8 Heterojunction for Water Purification: Enhanced Photocatalytic Performance and Mechanism"

_catalysts, doi:10.3390/catal13111450_

Round 1

Reviewer 1 Report

Comments and Suggestions for Authors

The MS is a very relevant piece of research and overall is well written, however I have the following minor suggestions for the authors:

A- Not all the achronims seem to be identified in the MS- I would suggest adding a list at the end with all of them properly defined. 

B- The section on Materials and Methods should be placed before Results

C-Figure 5 does not seem to show error bars-Please comment on this item

D-The description of Figure 7 could be improved to clearly identified  the degradation mechanisms by a proper label in the figure. 

Author Response

Reviewer #1: The MS is a very relevant piece of research and overall is well written, however I have the following minor suggestions for the authors.

  1. Not all the achronims seem to be identified in the MS- I would suggest adding a list at the end with all of them properly defined.  

Response: We are grateful to the reviewer’s thoughtful comments and we are really sorry for not providing detailed chromatogram and mass spectrometry. We have re-marked the chromatographic peaks and identified previously unlabeled peaks through the NIST spectrum library with GCMS (Line282-306). The re-labeled chromatogram and mass spectrometry data are presented in Figure S1 and Figure S2 in the supplementary information. In accordance with the suggestion, we have included a table listing the peaks identified through mass spectrometry (Table S1).

2.The section on Materials and Methods should be placed before Results.

Response: Thanks for the reviewer’s suggestion and we have carefully checked the format requirements of the journal. According to the template requirements, "Materials and methods" must come before "Results and discussion".

  1. Figure 5 does not seem to show error bars-Please comment on this item.

Response: We are grateful to the reviewer’s thoughtful suggestions. We employed Origin to compute the mean and standard deviation. However, when the degradation concentration(C) is divided by the original concentration(C0), the resulting ratio values are so close that they lead to very small error bars in the plot, , most of which are obscured by the data points (ICONS). Consequently, we chose not to include error bars when drawing the Figure 5.

  1. The description of Figure 7 could be improved to clearly identified the degradation mechanisms by a proper label in the figure.

Response: We are grateful to the reviewer’s thoughtful suggestions. We reanalyzed the previous mass spectrometry to identify the main degradation products (Line282-306). According to the MS data and previous report, we proposed a possible degradation pathway of BPA and draw an illustration of the degradation pathway (Figure 7). Additionally, we enhanced the understanding of the BPA degradation mechanism by providing clear labels on the degradation pathway diagram. Thanks.

Reviewer 2 Report

Comments and Suggestions for Authors

The authors have tackled the challenge of developing efficient photocatalytic technology for water purification. They've successfully created a novel WO3/SnIn4S8 (WSI) S-scheme heterojunction through in-situ precipitation and hydrothermal synthesis, allowing simultaneous removal of Bisphenol A (BPA) and reduction of Cr(VI) in contaminated water. The WSI-12% configuration has demonstrated significantly improved photocatalytic efficiency under visible light compared to pure WO3 and SIS, thanks to its ability to separate charge carriers and maintain strong photoredox capabilities. Radical-trapping experiments and ESR analysis have confirmed the role of superoxide and hydroxyl radicals in BPA degradation and Cr(VI) reduction, highlighting the potential of this research for complex pollutant removal in wastewater treatment.

This work has some concerns that should be rejected,

This work has previously been discussed in previous papers similar to this material that claims to be a Z-scheme heterojunction. So how authors have made sure that the material makes an S-Scheme heterojunction? 

In Figure 1, there are too many peaks shown unassigned, how authors are showing that heterojunction only as a result of the claimed heterostructure? how do they guarantee that the other components are not effective in the heterojunction?

Why have authors used diethyldithiocarbamate when there are cheaper S sources for synthesis?

In Figure 5b, there is a sharper slope for BPA degradation in the dark than light, how is it justifiable?

In Figure 7, several intermediates are impossible to produce. How possible?

ESR data are not so clear, how they produce DMPO-·O2- and DMPO-·OH intermediates? HOW BASELINES can be identical?? This is data manipulation!

Figure 9 is full of criticism and typos

Reviewer 3 Report

Comments and Suggestions for Authors

The work is interesting and has been done at a high scientific level. The conclusions of the work are sufficiently supported by the results obtained. At the same time, there are a few points left, clarification or disclosure of which will improve readers' understanding of the content of the work.

- Figure 1 and Figure 7 are too small to be readable

- Table 1 "Kinetic reaction rate constants for the degradation of BPA and reduction of Cr(VI)" present "reduction rate constants(k)/min-1" among with values for both BPA and Cr(VI). Which is true, reduction or degradation? 

- Figure 5d it is better to show y-axis like "BPA ln(C/C0)" to exclude all the possible misunderstandings

- It is necessary to specify the conditions for registering EIS data: which radiation is used, at what potential and at what intensity of radiation is recorded. It would be better to select a potential equal to open-circuit-potential because in this case there is no external separation of charges and exactly the same processes occur in the volume as in the photocatalytic process

- It is necessary to decide what is generally said in the work: about the processes of recombination or diffusion (lines 199-204) 1. The diffusion process should be described by the Gerischer impedance, then indeed the smaller the semicircle, the shorter the time: the shorter the time, the higher the rate for the charge carrier to exit the catalyst volume. 2. The recombination process, which should be described by an ellipse with a maximum position corresponding to the frequency of the recombination process. The characteristic period is T = RC = Z'/(2Pi Z")

- Initially, there was 50 ml of the solution, each time you took samples of 3 ml. 10 samples in 4 hours (according to the data provided) this is 30 ml of the taken solution, which greatly affects the volume of the processed system. Have you taken into account in any way such a change in the volume of the system?

- Please provide a scheme of the actual experimental setup from which it would be visible where and how the reactor lighting is organized so that it can be judged on the completeness of the use of incident lighting

Comments on the Quality of English Language

- missing "oxidation" in the very begining of the abstract (line 7)

- "Bisphenol" in line 11 and "Superoxide" in line 18 written with the capital B and S, what for?

Author Response

The work is interesting and has been done at a high scientific level. The conclusions of the work are sufficiently supported by the results obtained. At the same time, there are a few points left, clarification or disclosure of which will improve readers' understanding of the content of the work.

  1. Figure 1 and Figure 7 are too small to be readable.

Response: Thanks for the reviewer’s suggestion and we are sorry not to provide the figure with enough resolution. Figure 1 and Figure 7 have been replaced with enough resolution to meet the publication standards of the Journal.

  1. Table 1 "Kinetic reaction rate constants for the degradation of BPA and reduction of Cr(VI)" present "reduction rate constants(k)/min-1" among with values for both BPA and Cr(VI). Which is true, reduction or degradation?

Response: We are grateful to the reviewer’s thoughtful comments and we are sorry for bothering you with the unclear description. Despite different reaction mechanisms, the descriptions are unified as the photocatalytic degradation of BPA and the photocatalytic reduction of Cr(VI). (Line249-252).

  1. Figure 5d it is better to show y-axis like "BPA ln(C0/C)" to exclude all the possible misunderstandings.

Response: We are grateful to the reviewer’s thoughtful comments and we are sorry for bothering you with the unclear figure. Figure 5d has been replaced to eliminate possible misunderstandings.

  1. It is necessary to specify the conditions for registering EIS data: which radiation is used, at what potential and at what intensity of radiation is recorded. It would be better to select a potential equal to open-circuit-potential because in this case there is no external separation of charges and exactly the same processes occur in the volume as in the photocatalytic process.

Response: Thanks for the reviewer’s suggestion and we have added EIS measurement condition (Line402-404). EIS measurements were conducted using EIS system (CHI660, chinstruments, China). We performed EIS in the frequency range of 0.1–105Hz with the initial voltage of 0 V and amplitude 5mV.

  1. It is necessary to decide what is generally said in the work: about the processes of recombination or diffusion (lines 199-204) 1. The diffusion process should be described by the Gerischer impedance, then indeed the smaller the semicircle, the shorter the time: the shorter the time, the higher the rate for the charge carrier to exit the catalyst volume. 2. The recombination process, which should be described by an ellipse with a maximum position corresponding to the frequency of the recombination process. The characteristic period is T = RC = Z'/(2Pi Z").

Response: Thanks for the reviewer’s suggestion and we have revised the EIS discussion. The high-frequency EIS is used to assess the electron transfer capability of the sample. The rate of electron transfer is reflected by comparing the radius of the impedance graph: a smaller radius indicates faster electron migration. Based on the above principle, we conducted a re-analysis of the EIS spectrum data. (Line201-204).

  1. Initially, there was 50 ml of the solution, each time you took samples of 3 ml. 10 samples in 4 hours (according to the data provided) this is 30 ml of the taken solution, which greatly affects the volume of the processed system. Have you taken into account in any way such a change in the volume of the system?

Response: We are grateful to the reviewer’s thoughtful comments. During the photocatalytic experiment, most samples were collected at one-hour intervals, the sample used for hexavalent chromium analysis was 3ml, while the sample used for BPA analysis was about 1ml, so the total sample volume of approximately 10 mL. Additionally, the catalytic dose, BPA concentration, reaction volume, and other factors were proportionally reduced, ensuring that the volume change did not affect the experimental results.

  1. Please provide a scheme of the actual experimental setup from which it would be visible where and how the reactor lighting is organized so that it can be judged on the completeness of the use of incident lighting.

Response: Thanks for the reviewer’s suggestion. The photocatalytic experiments were performed in a double-walled quartz jacket filled with cool water under visible light irradiation. The distance between Xe lamp and reactor was adjusted to 10 cm to achieve an incident light intensity of 100 mW•cm-2 (equivalent to one sun). According to the suggestion, we have added experimental equipment as shown in Figure S3 in the supplementary information.

  1. missing "oxidation" in the very begining of the abstract (line 7). "Bisphenol" in line 11 and "Superoxide" in line 18 written with the capital B and S, what for?

Response: We are really sorry about this mistake that we could have avoided. Accordingly, Accordingly, we have re-checked the grammar throughout the paper and revised some miss spelling (Line7, Line11, Line18).

Round 2

Reviewer 2 Report

Comments and Suggestions for Authors

As I said before, this paper falls out of the scientific standards of research by some misclaims and serious errors on the basis of the scientific level. Again, as I said, this paper should be rejected.